# Oncostatin M in the Regulation of Connective Tissue Cells and Macrophages in Pulmonary Disease

**DOI:** 10.3390/biomedicines7040095

**Published:** 2019-12-05

**Authors:** Carl D. Richards, Fernando Botelho

**Affiliations:** McMaster Immunology Research Centre, Department of Pathology and Molecular Medicine, McMaster University, Hamilton, ON L8S 3Z5, Canada; botelhf@mcmaster.ca

**Keywords:** Oncostatin M, gp130, IL-6, cytokines, connective tissue, macrophages, lung inflammation, ECM remodeling

## Abstract

Oncostatin M (OSM), as one of the gp130/IL-6 family of cytokines, interacts with receptor complexes that include the gp130 signaling molecule and OSM receptor β OSMRβ chain subunits. OSMRβ chains are expressed relatively highly across a broad array of connective tissue (CT) cells of the lung, such as fibroblasts, smooth muscle cells, and epithelial cells, thus enabling robust responses to OSM, compared to other gp130 cytokines, in the regulation of extracellular matrix (ECM) remodeling and inflammation. OSMRβ chain expression in lung monocyte/macrophage populations is low, whereas other receptor subunits, such as that for IL-6, are present, enabling responses to IL-6. OSM is produced by macrophages and neutrophils, but not CT cells, indicating a dichotomy of OSM roles in macrophage verses CT cells in lung inflammatory disease. ECM remodeling and inflammation are components of a number of chronic lung diseases that show elevated levels of OSM. OSM-induced products of CT cells, such as MCP-1, IL-6, and PGE2 can modulate macrophage function, including the expression of OSM itself, indicating feedback loops that characterize Macrophage and CT cell interaction.

## 1. Introduction

Networks of cellular interactions are governed by many factors, including cell–cell contact, the presence of mediators (exogenous and endogenous) at sufficient concentrations, and the sufficient expression of membrane-bound receptors or soluble receptor complexes that enable cell responses. Mediators include low molecular weight endogenous compounds (such as small metabolites and prostanoids, as well as other larger polypeptides, such as alarmins, cytokines and chemokines) and their receptors to generate alterations of cell behavior. The gp130 family of cytokines are among many cytokines and other mediators that shift in levels and patterns of expression over the course of inflammatory processes due to tissue/organ injury, as a result of trauma, infection, autoimmune, allergic chronic disease, and cancer [1,2,3], and may have roles in metabolism involving adipose tissue [4]. The gp130 cytokines (subject of this special issue) have generated much study in many homeostatic and disease orientations, and of these the family member oncostatin M (OSM) has received increased interest in recent years. OSM is a 22–28 kD protein released by activated leukocytes that binds cell membrane receptors composed of gp130 and the OSMRβ chains [5,6]. Several reviews exist that explore the details of OSM functions in inflammation [7]; OSM and IL-31 receptors and signaling [6]; functions of OSM, focusing on stromal cells [8,9]; actions of OSM in the central nervous system [10]; and bone metabolism [11]. On the basis of the ability of OSM to markedly regulate connective tissue cell types (see below), there is recognition of an OSM–stromal cell axis, supported by our own work and others, that participates in homeostasis, immune responses, and disease. This axis has been recently updated comprehensively by West et al [8]. 

In context of pulmonary disease, OSM has been shown to be elevated in idiopathic pulmonary fibrosis [12], severe asthma [13], chronic rhino-sinusitis [14], and Chronic Obstructive Pulmonary Disease (COPD) [15]. However, the specific cause-and-effect relationship between OSM and each of these lung/airway conditions in vivo awaits further mechanistic investigation, as well as the effects of targeting the OSM pathway in preclinical and clinical interventions for lung diseases. Here, we will specifically focus on what is known about OSM and functions in connective tissue cells and macrophages of the lung. These cells are important in the control of acute and chronic lung inflammatory diseases, and we recognize that macrophages are only one cell type amongst many immune cells involved in homeostasis and either innate or adaptive immunity and chronic disease. For the purpose of this review, we refer to connective tissue cells rather generally as those cells that contribute to the formation and homeostasis of connective tissue structure, integrity, and function. We review OSM receptor expression, regulation of the extracellular matrix, OSM regulation of lung fibroblasts, epithelial cells, smooth muscle cells, and endothelial cells, all of which contribute to structure/function of the lung as a mucosal surface interacting with the environment. We also comment on potential sources of OSM in the lung, and its regulation in feedback loops by mediators derived from connective tissue cells. 

## 2. Oncostatin M Receptors and Cell Responses

The function of cell receptors for mediators requires receptor components to be adequately expressed by responding cells. For the gp130 cytokines, these receptors are primarily membrane-bound and are composed of multiple chains encoded by different genes [2,3,6]. The OSM ligand is released by activated T cells, macrophages and neutrophils, and interacts with receptor complexes composed of gp130 and the OSMRβ chain subunits in both human and mouse systems, termed the specific (or type II) OSM receptor complex (extensively reviewed recently [6]). These receptors engage several intracellular signaling cascades, including predominant activation of the STAT3 pathway; in the human system, it has been established that an OSM ligand can also interact with the leukemia inhibitory factor receptor (LIFR complex), composed of gp130 and LIFRα subunits, (also termed the Type I OSM receptor) [2,3,6]. Publicly available databases, such as the human protein atlas (https://www.proteinatlas.org/) provides a cursory view of the tissue and cell type expression of these receptors. The data summarized indicates that the receptor subunits for gp130 cytokines are differentially expressed in separate cell types, and that the gp130 and OSMRβ chains required for OSM-specific (Type II) receptors are well expressed on fibroblasts and connective tissue cells, whereas OSMRβ is expressed to a much lower/no level on monocytes/macrophages. This helps predict the potential function of OSM in networks in vitro and in vivo, with connective tissue cells as predominant responders to OSM, and monocyte/macrophages as low/non-responders.

Consistent with the receptor expression, OSM has been shown in vitro to regulate lung fibroblasts, lung epithelial cells, airway smooth muscle cells, and endothelial cells, all of which possess OSMR complexes in sufficient quantity to enable responses to the OSM ligand through its specific OSMRβ/gp130 (Type II) receptor complex. Although IL-31 interacts with a receptor complex that includes the OSMRβ chain, the other chain IL-31Rα is also required for IL-31 signaling (reviewed as above in [6]). We have observed low/no responses to IL-31 in lung fibroblasts or smooth muscle cells in-house, and thus have focused on OSMR complex function in our own work.

## 3. Extracellular Matrix Remodeling

Normal tissue differentiation and normal healing of adult damaged tissue requires appropriate regulation of ECM remodeling. In chronic lung inflammation and pathology, abnormal remodeling results in net ECM protein accumulation (such as that seen in pulmonary fibrosis or severe asthma) or net catabolism (such as that seen in emphysema). Pathological ECM remodeling is evident in liver fibrosis/cirrhosis, scleroderma, atopic dermatitis, psoriasis, and rheumatoid arthritis; in the latter, net cartilage catabolism, bone formation, and fibrosis is also evident. Studies have shown that OSM over-expression in animal models has marked effects in generating increased ECM/fibrosis in the lung, skin, and synovium [16,17,18,19]. 

In the lung and skin, connective tissue fibroblasts are principle sources of matrix proteins, including collagens, fibronectin, and laminin, as well as enzymes that cross link such components. Fibroblasts and macrophages are sources of enzymes and protein inhibitors that regulate the metabolism of ECM once it is laid down. Such catabolism involves the matrix metalloproteinase (MMP) family of enzymes and serine proteinases, such as elastase, that collectively have a broad spectrum of substrate specificities. Tissue inhibitors of metallo-proteinases (TIMPs1-4) inhibit MMP function, and the balance of MMP/TIMP levels has been a central paradigm in the tissue remodeling processes [20,21]. Many growth factors may participate in the regulation of ECM synthesis and its metabolism, and a prominent cytokine that has been well-studied in the induction of ECM deposition (fibrosis) is transforming growth factor-β (TGF-β). 

### 3.1. Transforming Growth Factor and Oncostatin M in Extracellular Matrix Remodeling In Vivo

Mechanistic studies of net ECM deposition implicate TGF-β activity through activation of its canonical signaling SMAD2/3 pathway [22,23]. The ECM can be markedly induced in rodent lung tissue upon over-expression of TGF-β [24]. However, although TGF-β modulates the MMP/TIMP balance and is thought to be a central driver of fibrosis in the lung [25], TGF-β-independent pathways have also been suggested in the lung, on the basis of data in animal models. In clinical trials targeting tyrosine kinase signaling of TGF-β, the drug Nintedanib showed slowing but not a stoppage of progression, and may generate severe side effects [26,27]. This also suggests that TGF-β-independent pathways of ECM remodeling contribute to progression. Another recently approved drug, Pirfenidone, although also an improvement to previous therapies, has shown limited efficacy in patients and side effects [28], and its mechanisms of action are not yet clear. Thus, further study and alternative interventions are merited, which may also have applications to other fibrotic conditions.

ECM deposition generated in a house dust mite model of airway allergic inflammation in mice was not reduced by TGFβ blockade [29], and our own studies show that OSM overexpression in mouse lungs induces fibrosis in a SMAD3-independent manner. Previous work has also suggested excess STAT-1/STAT3 signaling can markedly exacerbate Bleomycin-induced fibrosis in mouse lungs that does not require the canonical TGFβ-SMAD3 pathway [30]. Precise mechanisms or the roles of STAT3 activating cytokines (such as OSM or other gp130 cytokines) and macrophages (or other immune cells) in such pathways are not yet fully defined. In BALB/c mouse lungs, OSM induces marked pSTAT3 activation with concomitant pSMAD 1/5 suppression [18] and no detectable evidence of TGFβ –SMAD2 activation. Since pSMAD1/5/8 is part to the canonical BMP signaling pathway [31], which appears protective in lung fibrosis models [32], it is possible that over-activation of pSTAT3 mediates fibrosis indirectly through modulating the BMP/pSMAD1 pathway. Future work in exploring these observations and mechanisms in other systems may substantiate this speculation.

### 3.2. Oncostatin M Regulation of Connective Tissue Cells In Vitro 

Fibroblasts derived from several tissue sites, including lungs, synovium, skin, and cardiac tissue, respond robustly to OSM in tissue culture. Earlier studies showing collagen stimulation by OSM in skin fibroblasts [33] were followed up by studies examining fibroblasts derived from lung. Scaffoldi et al. found that OSM induces collagen production, proliferation, and reduced apoptosis in human lung fibroblasts [34]. OSM also induces TIMP-1 in fibroblasts [35], which impinges on the MMP/TIMP-1 balance and contribution to ECM remodeling. When lung fibroblasts were stimulated in vitro with OSM in combination with TNF, O’Kane et al. showed that MMP-1 and MMP-3 secretion increased synergistically, whereas that of TIMP-1 and TIMP-2 decreased [36], supporting the ECM remodeling functions of OSM in pulmonary diseases with elevated OSM and TNF, such as tuberculosis. Nagahama [37] has more recently shown that OSM directly induces human lung fibroblast migration, alpha- smooth muscle actin (αSMA), fibronectin, and TGFβ expression, albeit the magnitude of induction is relatively small. Studying myofibroblast generation in vitro is complex due to the drifting of fibroblasts toward myofibroblasts in plastic surface culture conditions. However, the elevation of αSMA and fibronectin by OSM implies the ability of OSM to directly induce lung myofibroblast phenotypes. The OSM effect on human lung fibroblasts appears to be dependent on the activation of STAT3, since pharmacological inhibition of STAT3 could repress the OSM-induced responses [37]. This is interesting in the context of the results of STAT3 over-activation and robust exacerbation of bleomycin-induced fibrosis in animal models that was not affected in SMAD3−/− crossed mice [30]. Both observations suggest that TGFβ-independent pathways to fibrosis can exist. Hepatocyte growth factor (HGF) is also induced by OSM in human lung fibroblasts, and JNK signaling pathway inhibition reduces its effect [38]. HGF has multiple functions, including mitogenesis of epithelial tissues, and has been shown to cooperate with TGFβ in mediating myofibroblast transformation of oral fibroblasts [39], and thus has roles in wound healing. 

Regulation by OSM of collagen genes (Col1A1, 3A1) at the mRNA level have been observed in mouse lung fibroblasts and NIH3T3 mouse fibroblasts, concomitant with prominent STAT3 activation in vitro [18,40,41]. Interestingly, STAT3 has also been implicated in fibroblast senescence, which may participate in the etiology of pulmonary fibrosis [42,43]; however, whether OSM as a STAT3-inducing cytokine activates senescence pathways directly is not yet clear. In contrast to these studies in lung fibroblasts, recent studies by Huguier et al. have shown that OSM counteracts TGFβ–induced effects in skin fibroblasts [44]. This may indicate heterogeneity in ECM-modulating gene control by OSM in fibroblasts derived from different organs.

## 4. Regulation of Inflammatory Mediators in Lung Connective Tissue Cells

### 4.1. Chemokines

Although OSM was initially studied as a cytokine regulating the proliferation of cancer cells, work suggesting its potential roles in inflammation was later indicated by its regulation of IL-6 [45], granulocyte-macrophage colony stimulating factor (GM-CSF) and granulocyte CSF (G-CSF) [46] in human umbilical cord endothelial cells, as well as the acute phase response in hepatocytes [47], followed by OSM regulation of chemokines in mouse systems [48]. In response to OSM, mouse lung fibroblasts express chemokines (chemokine ligands, CCL), such as eotaxin-1 (CCL-11), as well as vascular cellular adhesion 1 molecules VCAM-1 in vitro [40,48]. OSM also synergizes with IL-4 or IL-13 to further induce eotaxin-1/CCL-11 production. This facilitates inflammatory responses that involve the accumulation of eosinophils into the lung tissue. In the mouse, such accumulation requires the action of additional pathways in vivo, since eotaxin-1 protein levels and eosinophil accumulation in broncho-alveolar lavage fluid (BALF) were absent in STAT6-deficient mice, implicating dependency on IL-4 or IL-13 and their canonical STAT6 signaling pathway [17]. Human airway smooth muscle cells also express monocyte chemoattractant protein 1 (MCP-1/CCL-2) in synergistic fashion in response to OSM and IL-4 or IL-13 combinations [49]. CCL-2 is a potent chemo-attractant factor for monocytes, T cells, and dendritic cells. Expression of the neutrophil chemo-attractant IL-8 (CXCL8) is induced synergistically by OSM and IL-17A in human airway smooth muscle cells [49]. In models of acute bacterial infection with *E. coli* in mouse lungs, OSM is rapidly elevated and is required for the induction of CXCL5 and neutrophil recruitment [50]. The observation that OSM function can interact with Th2 skewed cytokine function to induce eotaxin-1 has also been found in airway smooth muscle cells. Faffe et al. showed that OSM induced eotaxin-1 (CCL-11) through a STAT-3 pathway, and acted in synergy with IL-4 or IL-13 [51]. The mechanism of synergy may well involve OSM induction of IL-4Rα chains on the surface of airway smooth muscle cells [51] and lung fibroblasts [40], rendering these cells more sensitive to lower concentrations of IL-4 or IL-13.

### 4.2. Interleukin-6

In lung fibroblasts and airway smooth muscle cells, OSM synergizes with IL-1 or IL-17A to induce IL-6 expression [49,52]. In vivo, OSM overexpression in mouse lungs induces significant levels of IL-6 protein found in the BALF. In IL-6 knockout (KO) mice, the inflammatory effects of overexpression of OSM, including eosinophil cell infiltration and chemokine levels, are largely ablated [53]. Thus, IL-6 is required for OSM-induced inflammatory effects in the lung. The IL-6 generated is likely derived from both connective tissue cells and incoming inflammatory cells. More recent studies have shown that overexpression of OSM induces the accumulation of alternatively activated (AA) macrophages, as defined in the mouse as Arginase-1+/CD206+ [54]. IL-6 is required for this effect, since AA accumulation is completely ablated in IL-6 deficient animals. Interestingly, overexpression of IL-6 alone is not sufficient to induce lung accumulation of these AA macrophage cell types [54], likely due to the additional requirement of AA macrophage-skewing cytokines IL-4/IL-13. In vitro, IL-6 potentiates the IL-4/IL-13-induced AA macrophage skewing towards a hyperpolarized AA macrophage phenotype [55]. Mauer et al. showed that this occurs through IL-6 up-regulation of the IL-4Rα on macrophages, enabling higher IL-4 signaling [56]. Such AA macrophages have been implicated in the induction of lung fibrosis in animal models. Other data have shown that PGE2 and IL-6 released by cervical cancer cells can induce skewing of macrophages to the AA phenotype [57], which have also been implicated as tumour-promoting cells.

### 4.3. Vascular Endothelial Growth Factor (VEGF) and Prostaglandin E 

OSM has been characterized as an angiogenic factor [58], and acts on vascular endothelial cells in a pro-inflammatory manner [59,60]. These studies were completed in aortic vascular endothelial cells, but whether pulmonary vasculature endothelial cells respond in the same manner is not clear at this time. OSM also synergizes with IL-1 or TNF in the regulation of VEGF [61] by airway smooth muscle cells, which may contribute to lung vascular alterations. OSM also synergizes with IL-1 in the up-regulation of cyclo-oxygenase-2 (COX-2) and PGE production by human vascular smooth muscle cells [62]. That OSM can synergize with IL-1 or TNF as pro-inflammatory cytokines has long been recognized in other systems, including articular cartilage chondrocyte cultures and cartilage degradation [63]. These actions in cartilage are mediated by a selective up-regulation of MMPs, such as collagenase-1, which result in the net degradation of collagen in articular cartilage in vitro and in vivo [62,63,64,65].

### 4.4. Lung Epithelial Cells and IL-33

The lung parenchyma is constructed to support the function of alveoli and capillary network for gas exchange. The intimal surface is lined with pulmonary endothelial cells, and luminal mucosal surface by epithelial cells, which change in populations (columnar, alveolar type 1, alveolar type 2) as one moves down the bronchial tree toward the terminal alveolar sacs. OSM was shown to regulate alveolar type II epithelial cells in vitro inducing α-1 proteinase inhibitor [66,67], whereas leukemia inhibitory factor (LIF) or IL-6 did not. This suggests that OSM acts on the specific type II OSM receptor in these cells, since LIF (functions at the LIFR or Type I OSM receptor) had little effect. A series of more recent studies in mouse systems have shown that OSM can regulate the alarmin IL-33 in mouse type II alveolar epithelial cells in vitro and in vivo [68], and induces liver endothelial cells to express IL-33 [69]. In the lung, columnar epithelial cell have been shown to express IL-33, but may not respond directly to OSM with IL-33 induction [68]. Since type II alveolar epithelial cells respond robustly to OSM in vitro, or to overexpression of OSM in vivo [68], whereas columnar epithelial cells did not, this may suggest STAT3 activation is not sufficient by itself to induce IL-33 in vivo. Further work is required to determine if OSM regulates IL-33 in human lung epithelial cells or pulmonary endothelial cells. OSMRβ is expressed by various intestinal epithelial cell lines, enabling STAT3 activation [70], but whether OSMRβ is expressed differentially in different types of human airway epithelial cells [71] would be of interest. 

OSM elevated pSTAT3 and cell-associated IL-33 in type II alveolar epithelial cells (primarily localized to the nucleus) [68], but did not alter other alarmins such as high mobility group box 1 (HMGB1) or IL-1α (our unpublished data). Although the IL-33 promoter (NCBI reference sequence AC_000041) contains several consensus STAT binding sites, suggesting direct transcriptional regulation by activated STAT3, this has yet to be confirmed in in functional studies. As an alarmin, IL-33 is released upon cell damage/death, signals through receptor (ST2+) cells to modulate inflammation, and can be inhibited by soluble ST-2 (sST-2), as reviewed by others [72,73,74]. Apoptotic cell death results in caspase cleavage of the IL-33 full length precursor to render it inactive. Necrosis does not involve caspases, and thus necrotic cells can release full-length IL-33 (active at the IL-33 receptor). Full-length IL-33 can be cleaved by other enzymes to yield more and less active mature forms, or is oxidized to an inactive form [75]. Elastase, protease3, and neutrophil cathepsin G cleave the IL-33 proform polypeptide at different sites than caspase, resulting in bioactive mature IL-33. IL-33 functions in adaptive immunity by activating IL-33 receptors on Th2 cells to generate IL-4/IL-13 production and influence Th2/AA macrophage inflammatory milieu. Thus, OSM (elevated in severe asthma) may participate in allergic lung adaptive immune mechanisms through its induction of IL-33. 

Innate lymphoid (ILC) cells provide additional pathways to innate immunity/inflammation independent of T cells and acquired immunity, and are also suggested to play roles in asthma [76,77]. The ILC-2 subtype is ST-2+, and is highly responsive to IL-33. Unlike adaptive immune Th2 cell products that require specific antigen/allergen stimulation, ILC-2 cells can release Th2 cytokines (IL-5, IL-13) in an antigen-independent manner. Studies in human asthmatic subjects identify some patients that are non-atopic, which could be driven by innate immune mechanisms such as IL-33 activation of ILC-2 cells without the requirement of allergen-specific Th2 cell activation and the production of IL-4/13. An OSM–IL-33 axis may be a pathway that is involved, and if identified, might lead to different intervention strategies in subpopulations of asthmatics. IL-33−/− or ST-2−/− mice have been shown to be protected in models of Bleomycin-induced lung fibrosis compared to wild types [78,79]. Whether IL-33 is required for OSM-induced ECM remodeling is not yet known.

## 5. Sources of Oncostatin M Ligands and Macrophages

Details of macrophages, their roles in immunity and disease, and their subpopulations have been described by recent and comprehensive reviews [80,81,82]. Classically activated macrophages play important roles in fighting bacterial and viral infections. Subpopulations of macrophages appear to play different roles, including pro-inflammatory, tissue repair, and anti-inflammatory functions [80]. Pro-inflammatory and anti-inflammatory/healing cell populations have been described in the mouse system as classically activated (M1) or alternatively activated (AA) M2 macrophages. However, this is a rather simple dichotomy, and other subpopulations have been suggested. Numerous cytokines, including gp130 cytokines OSM and IL-6, can be released by activated monocyte/macrophages stimulated by TLR ligands [83,84,85]. In our studies on bone marrow-derived macrophages, OSM expression is elevated in macrophages differentiated to either an M1 or an M2 phenotype. Prostaglandin E2 has been shown to stimulate OSM expression in macrophages and microglia [86], as well as OSM production by liver Kupffer cells [87]. OSM has also been shown to be induced in monocyte macrophages by complement component C5a [88] and thrombin [89], indicating that a variety of pro-inflammatory and homeostatic agents have the potential to contribute to OSM regulation in inflammatory diseases. Whether pro-inflammatory, tissue repair, or anti-inflammatory macrophages specifically in the lung release OSM differentially awaits careful delineation in both mouse and human systems.

Several subpopulations of macrophages have been identified in mouse lungs, including monocyte-derived alveolar macrophages and tissue-resident macrophages. Misharin et al. [90] have eloquently shown that in the model of Bleomycin-induced fibrosis, monocyte-derived alveolar macrophages contributed to fibrosis, whereas tissue resident macrophages did not. The monocyte-derived macrophages showed a distinct pro-fibrotic transcription profile that shared genes with human alveolar macrophages from fibrotic vs non-fibrotic lung tissues. In examining the publicly available deposited database from this work [90] (*GSE82158 dataset deposited in Gene expression Omnibus (GEO)), it is evident that OSM transcripts are expressed significantly by both populations of these macrophages in mouse. Transcript levels for IL-6Rα and gp130 were considerably higher than those for OSMRβ. This suggests that, at least in mouse lung, macrophages can produce OSM and have the capacity to respond to IL-6, but have much lower capability to respond to the OSM ligand through the OSMRβ/gp130 complex. LIF has been shown to regulate mouse peritoneal macrophage STAT3 signaling and decrease oxygen radical output [91], suggesting an anti-inflammatory role. In that study, peritoneal macrophages were shown to express gp130 and the LIFRα, indicating a functional LIF Receptor complex, although this does not necessarily apply to lung macrophages. It is not clear yet in human lung macrophages whether OSM can regulate responses through either the OSMR or LIFR, and this needs further experimentation to discern. 

Hypoxic environments may also influence macrophage function and steer tumour-associated macrophages (TAMs) to a tumour-promoting phenotype [92]. In a model of 4T1/BALB/c-syngenic mouse model of breast cancer, Tripathi et al. [93] have shown that OSM is increased in the tumour hypoxic environment. In vitro, and in the context of a tumour cell-conditioned medium, OSM drives a change from an M1 to an M2 phenotype of THP-1 cells or peripheral blood monocytes. This suggests that in certain conditions, blood monocytes may respond to OSM, although again, it is not clear if this was through the LIF receptor (Type I OSM receptor) or the type II, OSM-specific receptor. In macrophages derived from mouse adipose tissue, Minori et al. showed that OSMRβ was expressed in adipose macrophages, and that the mouse macrophage cell line RAW267 responded to OSM with a skewing toward AA or M2 macrophage phenotypes [94]. In contrast, Dubey et al. [54] found that mouse bone marrow-derived macrophages did not respond to OSM, consistent with low/non-detectable OSMRβ expression, whereas these cells did respond to IL-6 with STAT3 activation and enhancement of AA/M2 skewing by typical M2 stimuli IL-4/IL-13. In summary, various populations of macrophages can express OSM depending on the stimulus and tissue origin. Although there is evidence that certain macrophages themselves can respond to OSM, there is less evidence that this is through the specific Type II OSMR complex as opposed to the LIF receptor complex (Type 1 OSM Receptor).

### Other Sources of Oncostatin M

Clearly there are other potential sources of OSM from hematopoietic cells. These include, for example, Th1 cells [95] and activated T cells in skin inflammation [96]. CD8+ T cells from the lungs of subpopulations of scleroderma patients showed a pro-fibrotic gene expression pattern, including enhanced OSM and IL-4 [97]. Neutrophils are another source of OSM, originally identified by Grenier et al. [98] and subsequently by others in neutrophils in rheumatoid arthritis [99], skin wounds [100], bacterial pneumonia [50,101], and more recently, in neutrophils of the nasal mucosa [102] of patients with allergic disease. Oncostatin M reduced epithelial barrier function in air–liquid interface cultures, and was detected in the tissues of allergic asthmatic and chronic rhinosinusitis patients, as well as in biopsies from eosinophil eosophagitis patients [14]. These studies suggest a role for neutrophil-derived OSM in disrupting barrier function in mucosal diseases. Interestingly, in bacteria peritonitis infections, neutrophils are the main source of OSM, whereas peritoneal macrophages were not [103], suggesting tissue site-specific functions of macrophages. Local neutrophil infiltration may contribute to the OSM protein load at local sites of inflammation in other chronic pulmonary diseases. 

## 6. Feedback Loops between Connective Tissue Cells and Macrophages

As outlined in previous sections, OSM stimulation of lung connective tissue cells induces a broad array of responses. These include significant amounts of the chemokines MCP-1/CCL-2, IL-6, and PGE2. These mediators, in turn, will engage other pathways downstream of OSM production, including actions on macrophages themselves. MCP-1/CCL-2 is a potent chemo-attractant and activator of monocyte/macrophages, and PGE can potentiate OSM production by macrophages [86,87], indicating a positive feedback loop. IL-6 also activates macrophages through the IL-6Rα/gp130 complex to induce IL-4Rα expression, and this potentiates the skewing of these cells toward a hyperpolarized AA phenotype. The induction of IL-6 by OSM may thus indirectly contribute to ECM remodeling through such a pathway. These interactions are depicted schematically in Figure 1. Cell–cell contact between macrophages and fibroblasts in vivo may also add to the breadth/strength of responses induced. Lodyga et al. have shown that the co-culture of AA macrophages with lung fibroblasts induced the formation of myo-fibroblast phenotypes [104]. Interestingly, this required cell–cell contact [104] and was inhibited by a TGFβ blockade. In addition to cell–cell contact, cell–matrix contact may play an important part in OSM biology. Ryan et al. [105] have found that human OSM ligands bind to the ECM proteins collagens 1 and IX, laminin, and fibronectin under acidic conditions in vitro. Interestingly, OSM immobilized on such matrices retained biological activity, as assessed by subsequent co-culture with OSM-responsive breast tumour cell lines and STAT3 activation [105]. Whether biologically active OSM bound to the matrix is evident in inflammatory lung disease tissues in vivo is not clear at this time, and such investigations have strong merit. 

## 7. Targeting the Oncostatin M Pathway as a Potential Therapeutic

As noted previously, several lung conditions show elevated OSM including IPF, asthma, chronic rhino-sinusitis and COPD [12,13,14,15], each of which involve regulation of ECM remodeling in lung. Since the severity of ECM remodeling alters the normal function of these organs, the potential role of OSM and other gp130 cytokines in control of pathological ECM remodeling is of interest in defining potential new approaches to modify chronic disease outcomes. Differential roles of OSM in cartilage (catabolic) and bone (anabolic) accentuate the complexity of OSM roles in skeletal tissues. OSM roles in the lung may not have such a dichotomy of catabolic verses anabolic ECM remodeling. 

Targeting the OSM pathway can be accomplished by inhibiting the ligand or its receptor with neutralizing antibodies, or by inhibiting signal transduction with small molecular weight compounds. The first biologic assessed in patients (indication Rheumatoid Arthritis) was a monoclonal Ab to the OSM ligand (GSK315234), and while well-tolerated, resulted in a protein-carrier effect, increasing OSM half-life [106]. A newer, anti-OSM Ab (GSK2330811) is currently in trials for diffuse systemic sclerosis (ClinicalTrials.gov), and an antibody to the OSMRβ chain (KPL-716) is currently in trials for the skin condition prurigo nodularis (www.Kiniksa.com). The OSMRβ chain is also required for the IL-31 receptor complex, which has been clearly implicated in itch/pruritic skin conditions. Since KPL-716 targets the OSMRβ chain, it may serve a dual function in regulating both OSM and IL-31 driven processes. Information resulting from these trials as they progress will inform us about the potential for targeting OSM pathways in pulmonary diseases. 

## 8. Conclusions

Oncostatin M has a unique repertoire of cell targets in lung tissue, in the context of the IL-6 family of cytokines, due to differences in the abundance of OSMRβ expressed by separate cell types. Its action on the specific type II OSM receptor complex (OSMR/gp130) activates inflammatory and ECM remodeling pathways, and potentially contributes to the disease pathogenesis of multiple chronic pulmonary conditions. Feedback loops between connective tissue cell products and macrophage function contribute the OSM–cytokine network. The expression of the OSM ligand appears primarily restricted to hematopoietic cells, including macrophages, T cells, and neutrophils, and has been implicated in other tissue sites, including inflammatory joint and skin conditions. More recently, a role for OSM and the OSMRβ has been put forward for intestinal bowel disease patients who are refractory to TNF-blockade therapy [107]. Early-phase clinical trials currently exist that are targeting either the OSM ligand or the OSMRβ chain; however, the biology of OSM in inflammatory conditions still has many uncertainties to be unwound. These include the nature of the OSM protein and bioactive half-life when associated with ECM in disease, the precise nature of how and in what cells OSM is induced, and what factors are involved in the induction of the OSM gene in separate pulmonary diseases. Whether OSM would be useful as a diagnostic tool is hampered in part by its low/no detectable levels systemically in most conditions. This may reflect a normal homeostatic function within local tissue compartments and its apparent biological efficacy at low concentrations. In targeting a pathway for therapeutic use, one would need to consider the diverse effects of OSM across tissues, and may require selective tissue/organ targeting to minimize adverse effects. The OSM/OSM receptor pathways are indeed intriguing, and further exploration will generate potentially new approaches in treating chronic inflammation, including that found in pulmonary disease.

## Figures and Tables

**Figure 1 biomedicines-07-00095-f001:**
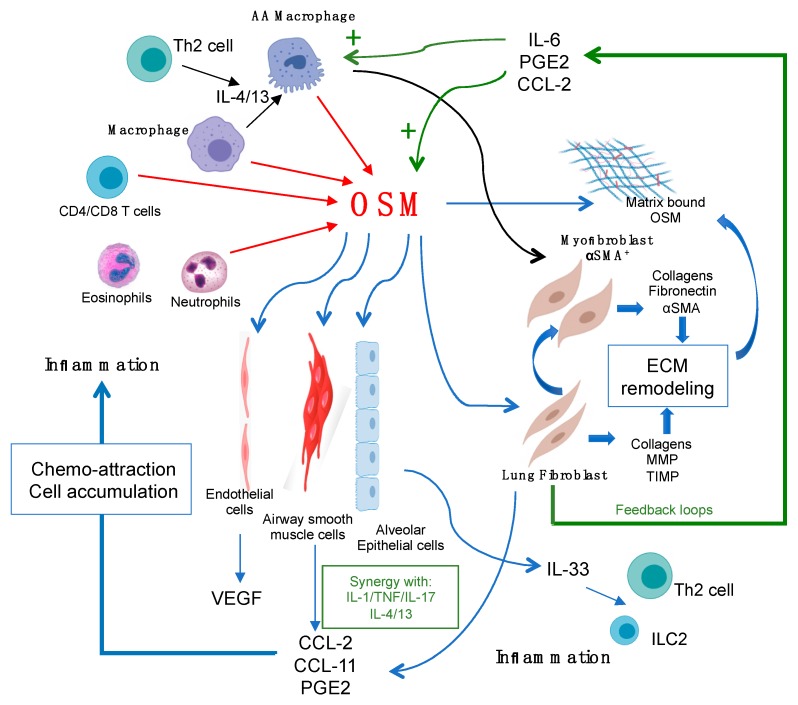
Oncostatin M (OSM) regulation of lung connective tissue cells, the extracellular matrix (ECM), and inflammation. OSM can be produced by a number of lung cell types (red arrows), such as macrophages, neutrophils, and T cells. OSM can act directly on target connective tissue cells of the lungs, including lung fibroblasts, smooth muscle cells, and endothelial cells (thin blue arrows) to produce the chemokines CCL2, CCL11, PGE2, and VEGF, as shown, which facilitates further inflammatory cell accumulation. OSM also stimulates epithelial cell overexpression of IL-33, which can stimulate Th2 cells and innate lymphoid (ILC)-2 cells once released. OSM also stimulates the accumulation of alternatively activated (AA) macrophages, which are thought to drive αSMA+ myofibroblast differentiation. Both lung fibroblasts and the myofibroblast production of matrix proteins results in increased ECM deposition. OSM ligand can also bind to the ECM, creating a bioactive reservoir of OSM. Production of IL-6, PGE2, and CCL2 by OSM-targeted cells provides feedback loops (green arrows) to further induce OSM production, macrophage activation, and hyperpolarization of AA macrophages (green plus symbols).

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
