# Peer review of "Oncostatin M in the Regulation of Connective Tissue Cells and Macrophages in Pulmonary Disease"

_biomedicines, 2019, doi:10.3390/biomedicines7040095_

Round 1

Reviewer 1 Report

Oncostatin M might play an important role in cell-cell crosstalk in lung micro environment. In the manuscript, authors  demonstrated related data and introduced related function of a couple  of cell types, such as macrophages and neutrophils, etc. According to the current title, related concept should be clarified. Pulmonary diseases refer to many disorders affecting function of lungs, such as asthma, COPD, pneumonia and tuberculous, lung cancer and other breathing issues. However, regarding to the current manuscript, authors did not introduce  data associated diseases, thus, authors should re-edit the manuscript according to  types of diseases, which help readers understand the cutting-edge research in above areas.

 The framework of this manuscript is not organized and clear. Authors listed details on related research, but it's difficult for readers to understand the trend of related research. Thus, authors  should re-organize the current manuscript.

Author Response

We agree that pulmonary diseases refer to a variety of disorders, however the published data of the precise roles of OSM in each of these human lung disease conditions is not yet clear. There is a relatively small amount of data published as yet on OSM in human Asthma and IPF, and less/none on OSM and tuberculosis, COPD or specifically lung cancer as diseases. To have the manuscript reorganized to specific lung diseases we believe would generate redundancy, since many of OSM’s predicted effects on specific stromal cells between diseases may be very similar. Instead, we organized the manuscript with a more reductionist approach and different lung cell types and responses.

The main message intended was the assessment of contribution of OSM to regulation of inflammation through lung stromal cells and macrophages, which in turn in part determines the nature of the lung disease pathology.

Thus, to increase the clarity of the introduction and intended focus of the review, we have considerably revised the introduction (lines 44-58), revised the section titles and reordered several sections for a more logical flow. The revised manuscript has been sent in track changes to easily see the modifications.

As others reviewers have suggested, we have also now incorporated an update on what’s know about human trials targeting the OSM pathways, and have incorporated an expansion of the unknowns and thus trends for the future in the conclusions

Reviewer 2 Report

The manuscript submitted for evaluation is a potentially valuable review paper concerning the issue of oncostatin M. Below I present my remarks to the manuscript, I make just a few comments:

All abbreviations should be explained in the first place where they appear, separately in the abstract and the main text. A significant initial fragment of the introduction was written without citation. This element requires completion also in other parts of the work. The introduction contains a lot of general information and poorly introduces the presented issue, this problem also applies to other parts of the work, where the authors in some situations limit themselves to general entry, without indicating / explaining the mechanism that was described in the cited work.

Author Response

Abbreviations are now explained appropriately.

The initial piece of the introduction is now written with more citations where we acknowledge they were lacking (eg lines 34, 38, 62,65/69). 

The introduction is now revised and restructured to clarify the presented issues. (please see response to reviewer 1)

Expansion of details of mechanisms and incorporated additional references are now added to various locations ( eg please see lines 122-127, 127-129 , 240-242, 262-264, new section 7.0). The revised manuscript has been sent in track changes to easily see these modifications.

Reviewer 3 Report

In this manuscript, Richards and Botelho reviewed several aspects of Oncostatin M in the connective tissue remodeling and their regulation of macrophages in chronic lung diseases. The review is well written, and several aspects of ECM remodeling and macrophage mediated inflammation in the context of OSM/OSMR are discussed.

Some of the aspects discussed in the current review looked redundant with Richards own review in ISRN Inflammation 2013. I would suggest discussing more about the relevant clinical studies till date with pharmacokinetics and pharmacodynamics of an anti-oncostatin M monoclonal antibody and their potential uses in pulmonary diseases. The only figure provided is not very helpful to focus on the message conveyed in the current review. An additional figure depicting the role of Oncostatin M in known diseases/disorders and their potential connection with cancer and chronic lung inflammatory diseases will help readers to stay focused.

Minor comments

Numbers given to headings and subheadings are out of place. Provide reference for line 199-200. The concluding remarks, although very precisely written, should also highlight some of the crucial unanswered questions in the field, so as to direct the readers to the main areas to be focused in the future research in the field.

Author Response

Heading numbers are now corrected and revised. We have reorganized the manuscript in order to clarify the focus and to create a more logical flow for readers. Please see response to reviewer 1.

The missing reference referred to has been incorporated into the revision.

A new section is added that briefly notes the current clinical trials with anti-OSM biologics ongoing for other conditions.

Important unanswered questions are expanded upon on in the conclusions as suggested. The revised manuscript has been sent in track changes to easily see the modifications.

Round 2

Reviewer 1 Report

Authors answered all concerns and improved the quality of manuscript.